# The Potential of Wine Lees as a Fat Substitute for Muffin Formulations

**DOI:** 10.3390/foods12132584

**Published:** 2023-07-02

**Authors:** Federico Bianchi, Mariasole Cervini, Gianluca Giuberti, Barbara Simonato

**Affiliations:** 1Department of Biotechnology, University of Verona, 37134 Verona, Italy; federico.bianchi_02@univr.it; 2Department for Sustainable Food Process, Università Cattolica del Sacro Cuore, 29122 Piacenza, Italy; mariasole.cervini@unicatt.it (M.C.); gianluca.giuberti@unicatt.it (G.G.)

**Keywords:** fat replacers, circular economy, wine lees, sustainable food products, rheology, sensory analyses

## Abstract

The current study evaluates the prospect of wine lees (WL), a costless by-product from Amarone winemaking, as a fat replacer in muffin formulation. WL have elsewhere replaced sunflower oil, allowing the creation of 0, 25, 50, 75, and 100% fat-substituted muffins named ML0, ML25, ML50, ML75, and ML100, respectively. Batter rheology, in addition to the textural and colorimetric characteristics, the pore dimension, and the sensory aspect of the different formulations were evaluated. The batter consistency (K) of fat-replaced muffins was lower than that of the control, while the hardness and chewiness of the end products were higher. ML25 and ML50 samples reached the highest volume, while the baking loss decreased due to WL’s fiber components. ML25, ML50, ML75, and ML100 accounted for caloric reductions of 9, 18, 22, and 26%, respectively, compared to full-fat muffins. Muffins with WL showed a darker crust and crumb as lightness (*L**) decreased. Moreover, *a** parameter increased with the increment of WL in the formulation, leading to a redder and less yellow-hued fat-replaced muffin. In conclusion, WL could effectively replace fat in the 25–50% range in muffins, achieving a final product with reduced calories, a higher dietary fiber content, higher volume, and promising sensory aspects.

## 1. Introduction

Non-communicable diseases (NCDs), such as obesity and type 2 diabetes, are related to an unbalanced diet [1,2,3]. Therefore, consumers have become aware of the issue and changed their food habits by reducing sugar and fat consumption while increasing their dietary fiber intake. On the other hand, developing novel and healthier food products is a real challenge for bakers and food companies. In bakery goods, fats act as a texture improver, flavor retainer, and moistener, influencing the mouthfeel and delaying starch retrogradation [4]. Moreover, shortening/oils in cake or muffin formulations tend to plasticize, lubricate, and strengthen the dough, improve gas retention, and create higher volume and softness in final products. Even though fat is an essential ingredient in several bakery formulas, reducing fat intake helps to reduce body weight and the risk of NCDs [5].

Indeed, fat replacers have recently undergone significant development, driven by consumers’ demand for low-fat foods [6]. Fat replacers can help to replace the whole or part of the fat within a food while maintaining the same functionality. The choice of the proper fat replacer is crucial and depends on the baked product’s formulation. Fat replacers can be carbohydrate, protein, or lipid-based and impart fat-like properties such as texturization, stabilization, emulsification, gelling, and thickening. Many fat replacers have been proposed in the last decade, and several are available in the market [6].

Use of the discarded grape by-product, such as grape pomace and lees from the fermentation process, could promote a circular economy and responsible use of natural resources, and also prevent environmental degradation, following the principles of the ONU Agenda 2030 [7]. Moreover, the upcycling of agro-industrial by-products could represent an opportunity to exploit since they are inexpensive and rich in dietary fiber and bioactive compounds. The effect of wine by-product addition in various cereal-based foodstuffs has been widely reported [8,9,10,11]. Wine lees (WL) are one of the main by-products of vinification. According to Bonamente et al. (2015), grapes’ by-products are comprised approximately 60% of pomace, 14% of stalks, and 25% of lees [12]. The WL are characterized mainly by mannoproteins from the yeast cell wall, mineral compounds, and dietary fibers, including pectin [13]. Pectin is a soluble fiber found in plants’ cell walls that is commonly used in several food formulas to control their textures and rheologies. Lim et al. (2014) reported that making cake with 30% of its shortening replaced by pectin resulted in batter with higher viscosity and cakes with a slightly lower volume than the control [14]. In addition, mannoproteins have already been tested as effective emulsifiers in different food formulations [15,16]. Emulsifiers, mixes of emulsifiers, or emulsifiers blended with dietary fibers or other food components could represent an effective fat replacer [4,17,18]. For instance, emulsifiers can compensate for the volume losses incurred due to fat replacement in cakes [14,15]. Moreover, several authors have reported discrete success in replacing fat with vegetable puree in bakery products [19,20,21,22]. Therefore, WL could be a promising fat-mimetic ingredient used to produce low-fat muffins. To our knowledge, no studies on WL as a fat replacer in foods are present in the literature. This work aimed to evaluate the technological, sensory, and nutritional characteristics of muffins formulated with increasing levels of WL (25, 50, 75, and 100%) as a fat replacer.

## 2. Materials and Methods

### 2.1. Ingredient and Preparation

Wheat flour was supplied from Macinazione Lendinara (Arcole, Italy), and sunflower oil, salt, sugar, fresh eggs, and baking powder were commercially available. The wheat flour’s composition was: total carbohydrates 71 g/100 g, protein 11 g/100 g, fat 1.2 g/100 g, and total dietary fiber 2.3 g/100 g (as reported on the label). The WL were obtained after the production of Amarone wine in Valpolicella and kindly provided by Cantina Sperimentale of Verona University (Verona, Italy). The WL were stored in plastic bottles in a −18 °C refrigerator and heated up to 25 °C upon use for WL characterization or muffin preparation. The muffins were prepared according to the formulation reported in Table 1. Briefly, wheat flour, salt, sugars, and baking powder were put in a planetary mixer set at medium speed for 5 min with tap water, sunflower oil, and whole eggs to obtain control sample ML0. Oil was replaced with WL in muffin preparation by 25, 50, 75, and 100% of its weight; samples were named ML25, ML50, ML75, and ML100, respectively. Fifty grams of muffin batter was poured into a pre-weighted silicone mold and then cooked at 180 °C for 20 min in a ventilated oven. Each formulation was prepared in triplicate.

### 2.2. Proximate Composition of Muffins and WL

The proximate composition determination for the muffin and WL included DM (method 930.15), ash (method 942.05), total sugars (total of D-glucose, D-fructose, and sucrose; Megazyme cat. no. K-SUFRG), crude protein (method 976.05), crude lipid (method 954.02), and total starch (method 996.11), using thermostable amylase (Megazyme cat. no. E-BSTAA), amyloglucosidase (Megazyme cat. no. E-AMGDF), and tartaric acids (Tartaric acid assay kit, Megazyme). The total, soluble and insoluble dietary fiber contents (TDF, SDF, and IDF, respectively) were determined according to method 991.43. [23]. Analyses were conducted in triplicate. In addition, WL was assessed for Brix, pH, and total phenol content (TPC). The TPC was determined according to Rainero et al. (2022) and expressed as gallic acid equivalent on DM, based on a 7-point standard curve [9]. The caloric contents of the muffin samples were calculated with the following equation:(1)$$\;\mathrm{Calories}\;\left(\frac{\mathrm{kcal}}{100\;g\;}\;\right)=9\times F+4\times C+4\times P$$
where F, C, and P represent the total amount of fat, carbohydrates, and protein in 100 g of muffin sample.

### 2.3. Rheology of Batters

A rheometer (DRS500 CP4000 PLUS, Lamy Rheology, France) equipped with an MSDIN-11 system was used for rheological analyses. Fifteen ml of batter were poured into the vessel and let rest for 1 min to reach a constant temperature of 25 °C. Shear stress was a function of shear rate over the 10–300 s^−1^ range. The results were fitted to the Power Law model Equation (2) by the software program Rheotex (Lamy Rheology, France).
(2)$$\;\tau =K\cdot \gamma^{n}$$
where “τ” is shear stress (Pa), “K” is the consistency index (Pa), and “n” is the flow index. All measurements were in triplicate.

### 2.4. Technological Properties

The moisture content of the muffins was evaluated using AACC method 44-15A [24]. To determine the baking loss, the mass difference between the batter and the baked muffins was measured. The volume was determined using rapeseed displacement method AACC 10-05.01 (AACC, 2000), and the density was calculated by dividing the weight of the sample by its volume. The heights of the muffins were measured using a digital caliper [10].

### 2.5. Texture Profile Analyses

Once the muffins had cooled down, their texture profile analysis (TPA) was conducted using a texture analyzer TX700 (Lamy Rheology, Champagne-au-Mont-d’Or, France) with a flat probe measuring 50 mm in diameter. For the TPA analyses, two cm of each muffin was cut from the bottom and the upper part was discarded. Seven measurements were performed for each batch. Hardness, cohesiveness, chewiness, and resilience values were collected. Hardness was calculated as the maximum force of the 1st cycle of compression, cohesiveness was the ratio of the work of the first compression and the second, and chewiness was a product of hardness, cohesiveness, and springiness, while resilience was the ratio between the time to reach the maximum peak force of the second cycle and the time of the first cycle.

### 2.6. Physical Characterization of Muffins

Muffins were cut to obtain a slice of 1–2 cm in height and about 20 cm^2^ in area. The scanner captured images of the muffin sections at a resolution of 600 DPI. Images were first converted to greyscale and binarized with ImageJ (version 1.8.0) to obtain the pore area fraction; the density of pores; the average pore diameter; D1,0 and D3,2; the pore perimeter; and the pore circularity [10]. Analyses were performed on five samples for each formulation.

### 2.7. Color Analysis

The color was measured by a reflectance colorimeter (Illuminant D65) (Minolta Chroma meter CR-300, Osaka, Japan) based on the color system CIE—*L** *a** *b**. Measurements were taken at five different points within the crumb and crust areas.

### 2.8. Sensory Analysis of Muffins

A quantitative descriptive sensory analysis (QDA) was conducted in the sensory test room of the University of Verona; the QDA was designed according to UNI ISO 8589 standards. The panelists were carefully selected based on their sensory sensitivity and ability to describe and communicate sensory perceptions. The panelists, consisting of 18 people (10 females and 8 males) between the ages of 22 and 28, recruited from the staff and students of the Department of Biotechnology at the University of Verona, were trained for 12 sessions of one hour each to evaluate the specific sensory attributes of muffins.

The panelists generated 10 sensory terms and were qualified to identify them. The following sensory attributes were evaluated: color intensity, egg taste, wine taste, wine odor, sweetness, acidity, chewiness, elasticity, hardness, and astringency. All sensory attribute definitions are reported in Table A1 (Appendix A). A nine-point scale, where 1 and 9 represented the lowest and highest intensities, respectively, was used [25]. The muffins were presented to panelists on covered plates, coded in a randomized and balanced manner, and then provided to the judges for the sensory analysis session. We asked the judges to rinse their mouths with water between each sample taste test. At the end of the session, the formats were taken, viewed, and statistically processed.

### 2.9. Statistical Analysis

All data are presented as mean values ± standard deviation, based on at least three measurements (n = 3). Analysis of variance (ANOVA) followed by a post-hoc Tukey test at a significance level of *p* < 0.05 was employed for comparing the means. Pearson’s correlation tests and statistical analyses were performed using XLSTAT Premium Version (2021.1.1, Addinsoft SARL, Paris, France) software.

## 3. Results and Discussion

### 3.1. Proximate Composition of Wine Lees

The chemical composition of WL is reported in Table 2. The current composition of WL from Amarone production had content values (DM basis) of polyphenol, tartaric acid, ashes, pH, and dietary fiber content similar to the WL from Bustamante et al. (2008) and Gòmez et al. (2004) [26,27]. However, in Amarone WL, a higher content of proteins and a lower content of lipids were detected. The peculiar grapevine and unique vinification method of Amarone wine (such as the juice extraction from withered grapes) could explain these differences [28].

### 3.2. Rheological Properties of Muffins’ Batter

Batter rheological data are reported in Table 3. All samples fitted the power law model equation, showing an r^2^ higher than 0.997. All batter formulations showed a shear-thinning (pseudoplastic) behavior (n < 1) over the studied shear rate (10–300 s^−1^), suggesting that the apparent viscosity decreased with the increase in shear rate. Bianchi et al. (2022) reported a similar performance in muffin batters made with grape pomace powder [10]. In the current study, the maximum flow index (n) value occurred in ML25 and ML50 samples, sequentially. In addition, ML75 and ML100 showed lower n than did the ML0, ML25, and ML50 samples, suggesting that their apparent viscosity increased less distinctly when the shear rate increased.

ML25, ML50, and ML75 showed similar consistencies, but were all lower than the full-fat sample, while ML100 had a higher consistency, resembling the behavior of 100% pectin as a fat replacer in cakes, as reported by Psimouli & Oreopoulou et al. (2012) [29]. The consistency increase is possibly due to the high thickening capacity of pectins from WL, since WL contains about 20% DM of dietary fibers, mainly pectins [30]. However, ML25, ML50, and ML75 have almost 35% less consistency than does the full-fat sample, which could be attributed to mannoproteins contained in WL [13]. Indeed, emulsifiers could lower batter consistency and apparent viscosity [4,31].

The higher consistency indicated a higher ability to entrap air during the beating process, resulting in a lower density of batters and lower volume development. If the K value is too low, cake or muffin volumes shrink because batters lose effectiveness in retaining air bubbles [32,33]. This can explain the greater volume of the control muffins compared to ML75 and ML100. Indeed, ML0 had higher consistency and higher volume. However, ML25 and ML50 showed the opposite results. The flow index n was negatively correlated with fiber, hardness, and chewiness, whereas no relationships were found between consistency and textural attributes or fiber content (Table A2).

### 3.3. Proximate Composition of Muffins

The chemical composition of the samples is reported in Table 4. The protein and total carbohydrate content remained unchanged between the different formulations. However, the ash content increased with increasing levels of WL in the recipe. This increase is related to the presence of macro- and micro-elements such as P, K, Mn, Zn, Ca, and Mg in WL [28]. The TDF content significantly increased from 0.30 g/100 g in the control sample to 3.16 g/100 g in ML100. This increment is due to the dietary fiber fraction of Amarone WL. As a result, ML100 can benefit from the claim of "source of fiber" as it contains more than 3 g/100 g of TDF [34]. Other studies have reported similar findings for bread, breadsticks, cookies, and muffins fortified with grape by-products [9,35,36,37,38]. Muffin fat content was successfully reduced from 13.14% for sample ML0 to 1.82% in sample ML100. By replacing fat, the fat-reduced muffins achieved lower calories per gram, as fat is the most calorie-dense nutrient in all formulations. Accordingly, muffins ML25, ML50, ML75, and ML100 accounted for calorie reductions of 9, 18, 22, and 26%, respectively, as compared to ML0. The TDF and ash content were positively correlated with the density of muffins and negatively correlated with the fat content, due to the substitution of fat with the Amarone WL (Table A2).

### 3.4. Technological Properties of Muffins

The oil replacement in muffins influenced most of the technological parameters, as depicted in Table 5. Even though the heights of the muffins decreased with oil replacement in the formulations, the detected differences were low and perceptible mainly in ML0 and ML25 compared to ML100. Fat replacement slightly affected sample volume: ML25 and M50 showed the highest, and ML100 accounted for the lowest volume. Moreover, a small increase in muffin density was revealed in LM75 and LM100, showing a WL impact in the batter density, probably due to the water adsorption capacity and the high fiber content.

In addition, the greatest volume was recorded for ML25 and ML50, increasing by 8.32 and 4.3% compared to the control, respectively, suggesting that the optimal substitution of fat with WL to achieve a better muffin volume should not exceed 50%. Higher fat replacement provoked a volume decrease of about 16% compared to the control. This could reflect the role of WL in capturing air during the beating step of batter preparation, stabilizing the air cell, and preventing their coalescence during the cooking, but only below the 50% threshold of replacement [22]. Even though parameters such K index should suggest a thicker batter and better gas retention ability, ML75 and ML 100 volumes were much lower than ML0, which was probably caused by an insufficient amount or the complete absence of oil in the formulations. Such conditions could promote bubble coalescence or a lack of gas retention and bubble expansion during cooking. For instance, Marchetti et al. (2018) reported that a low batter consistency led to cakes with lower volume [32]. The effect of dietary fibers on muffin volume has been widely reported. For instance, muffins have been enriched with brewer’s spent grain, peach fibers, linseed, flaxseed, wheatgrass, and grape pomace [10,39,40,41,42].

ML0 accounted for the highest baking loss in the study, whereas ML100 and ML75 showed the lowest ones. Bender et al. (2017) argued that the hydration capacity of the fiber fractions in fortified muffins led to a decrease in the baking loss when Riesling skin or Tannat skin powder was added [38]. The presence of fiber components in the WL could affect the baking performances of the muffins even if no correlation was detectable. Indeed, the correlation analyses (Table A2) showed that fiber content was negatively correlated with the volume. The moisture content increased in the fat-replaced muffins due to water in the WL. High moisture was detected by other authors using vegetal-based fat replacers in baked goods [22,43]. However, the increase in the moisture content of a WL-containing muffin could presumably have a negative impact on its shelf-life. Consequently, further studies are warranted to explore the relationship between WL inclusion levels and the microbial stability of related food products during their shelf-life.

### 3.5. Image Analyses of Muffins

The image analyses showed that the muffin had no significative differences in pore area fraction and pore circularity (Table 6). Indeed, Abdul Manaf et al. (2017) obtained similar results replacing fat with *Persea* puree [20], while Rodríguez-García et al. (2012) showed how inulin increased pores’ circularity and total cell area [44].

The ML25, ML50, and ML75 samples had higher pore densities than did the control or the ML100 sample. These data, for WL replacement < 100%, are in line with previous studies reporting that muffins with defatted sunflower seed flour, pecan nut expeller, and grape pomace powder had higher pore densities than did their controls [10,32,45]. Marchetti et al. (2018) stated that a higher pecan nut by-product amount promotes pore density in fortified muffins and a better gas retention capacity.

The pore perimeter values D1,0 and D3,2 showed a similar trend when replacing fat with Amarone WL. Indeed, ML100 led to a pore density similar to ML0, and thus to a more open crumb with a lower volume.

### 3.6. Color of Muffins

Consumers typically attribute darker crusts and crumbs to whole-grain baked goods. The addition of WL significantly changed (*p* < 0.05) the color of the muffins (Table 7). The color of the crust is formed during the baking process due to the caramelization of sugars and the Maillard reaction. Incorporating WL increases the Maillard reaction by increasing muffins’ amino acid and carbohydrate contents. Additionally, the crumb color is affected by cake ingredients, and the pigments of WL increase the color parameters of muffins [15,36]. The *L* and *b** parameters decreased (*p* < 0.05), both in crust and crumb, following the substitution level of the fat with WL, while the *a** parameter significantly increased (*p* < 0.05). Bianchi et al. (2022), Rainero et al. (2022), and Acun et al. (2014) obtained similar results, respectively, in muffins with distilled grape pomace powder, breadsticks with grape pomace powder, and cookies fortified with different grape products [9,10,46]. Moreover, the color parameter of the bread and cake fortified with grape pomace from winemaking showed a similar trend [37,47,48].

### 3.7. Texture of Muffins

As depicted in Table 8, hardness increased as WL replaced the oil in the formulation. For instance, hardness in the ML100 was 2.6-fold higher than that in the M0. Arifin et al. (2019) explored the effect of pumpkin puree as a fat replacer in muffins [21]. The pumpkin-puree-added muffin had higher hardness than the full-fat sample because lipids interact with gluten fibrils, promoting a softer muffin. Moreover, our findings agree with Grigelmo-Miguel et al. (2001) and Bianchi et al. (2022), as they reported that fiber-increased muffins had higher hardness than their respective control samples [10,40]. ML25 was the only replaced muffin with no significant difference compared to control ML0. Similarly, Hussien et al. (2016) observed no differences in 25% fat-replaced cantaloupe and squash puree muffins [22].

Cohesiveness increased with the rate of fat replacement level. These results agree with Rodriguez-Garcia et al. (2012), who stated that this increase might be due to a denser crumb cell structure in fat-replaced cakes [44]. Indeed, below the 75% level of fat replacement with WL there are positive correlations with pore density, perimeter of cells, D3,2, and specific volume. Grigelmo-Miguel et al. (2001) reported that peach fibers, when used as fat replacers, had little effect on either cohesiveness or springiness in muffins. However, the fat replacement was lower than 10%, and this reported effect could depend on the fat replacer concentration [40].

Chewiness values for fat-replaced samples were statistically higher than that of the control sample. Martínez-Cervera et al. (2011) obtained the opposite results in a muffin with cocoa fibers used as a fat substitute [49]. However, cocoa fiber as a fat replacer still contained fat and had a higher fraction of soluble dietary fibers, which could promote softness in bakery products [4]. The chewiness and hardness were positively correlated with the density of muffins, the ash, and the fiber content, while the flow index “n” and the fat content were negatively correlated (Table A2).

In the present study, all the samples had resilience values lower than 0.6, reflecting the presence of softening ingredients, such as sugars and fat, in the formulations. Indeed, resilience increased in muffins with lower fat content, underlying their role as tenderizers in bakery products.

Overall, hardness, cohesiveness, chewiness, and resilience increased with the incremental increase of WL used as a fat replacer in muffins.

### 3.8. Sensory Analysis of Selected Muffins

The replacement of oil with WL significantly influenced most of the selected sensory attributes, as depicted in Figure 1 and Table A2.

Color intensity changed with the rate of fat replacement in muffins and resulted in higher values for the ML100. Similarly, Bianchi et al. (2022) reported an increase in the perceived color of muffins fortified with distilled grape pomace powder, due to grape pigments. The WL’s inclusion affected almost all aroma- and taste-based sensory attributes (*p* < 0.05). The WL substitution reduced the egg taste perception and raised the wine taste. Noticeably, the egg perception was strongly affected by the WL addition, decreasing from a score of 7.9 for ML0 to less than 4.8 for the fat-reduced muffin. Wine odor and acidity were influenced by the oil replacement with WL as well. Bianchi et al. (2022) reported a similar effect in muffins fortified with grape pomace powder. The chewiness and hardness increased in muffins with the increased use of fat replacer. Indeed, the chewiness and hardness attributes of ML100 and ML75 were higher and statistically significant (Table A3) compared to ML0. For all samples, stringency and sweetness were not statistically different.

## 4. Conclusions

Reducing dietary fat in cereal-based food products represents a key focus of food industries. This study showed the potential of Amarone wine lees (WL) as a sustainable and effective ingredient used to partially replace fat ingredients in muffin formulations. Overall, nutritional, technological, and sensorial analyses indicated that the suggested amount of Amarone WL as a fat replacer in wheat-based muffins was between 25% and 50%, causing a calorie reduction and a higher dietary fiber content of the products without drastically compromising technological and sensorial aspects.

However, further optimization of the muffin formulas with 75% and 100% of WL by the addition of ingredients or alternative mixing and whipping strategies may improve their texture and sensory properties. Overall, this study highlights the potential of using WL as a sustainable fat-substitute in baked goods. Further research is also needed to explore the potential of WL as fat-replacer in different cereal-based food formulations.

## Figures and Tables

**Figure 1 foods-12-02584-f001:**
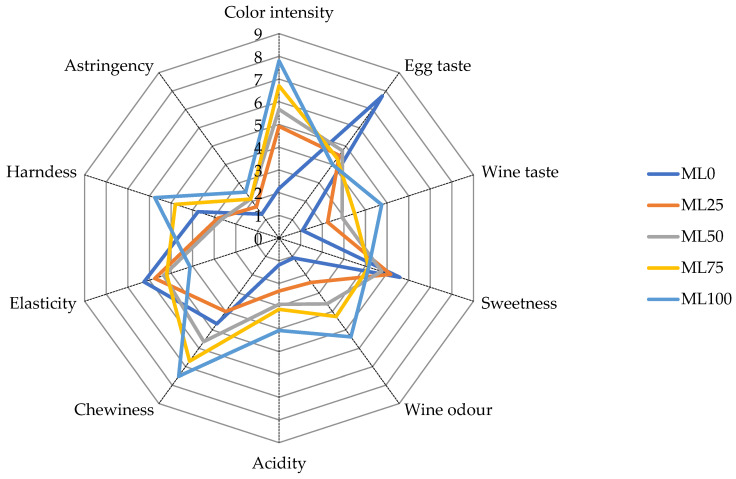
Spider plot of the sensory attributes of the muffin samples. ML0: control sample with no oil substitution; ML25: sample with 25% WL and 75% oil; ML50: sample with 50% WL and 50% oil; ML75: sample with 75% WL and 25% oil; ML100: sample with 100% WL.

**Table 1 foods-12-02584-t001:** Experimental formulas of control and fat-replaced muffin samples with WL.

Sample	Sugar (g)	Sunflower Oil (g)	Wine Lees (g)	Salt (g)	Baking Powder (g)	Wheat Flour (g)	Water (g)	Fresh Eggs (g)
ML0	50	25	0	1	2	100	50	35
ML25	50	18.75	6.25	1	2	100	50	35
ML50	50	12.5	12.5	1	2	100	50	35
ML75	50	6.25	18.75	1	2	100	50	35
ML100	50	0	25	1	2	100	50	35

**Table 2 foods-12-02584-t002:** Proximate composition, pH, Brix, and moisture content of Amarone wine lees. Protein, fat carbohydrates, ash, dietary fibers, phenol content and tartaric acid are reported on a DM basis.

Protein	Fat	Carbohydrates	Ash	Dietary Fibers	Phenols	Tartaric Acid	pH	Brix	Moisture
21.2 ± 3.4	0.07 ± 0	4.92 ± 0.11	10.27 ± 0.05	20.1 ± 3.0	1.53 ± 0.11	24.6 ± 1.2	3.41 ± 0.01	11.02 ± 0.21	87 ± 0.1

**Table 3 foods-12-02584-t003:** Reported values for batter flow index n and batter consistency index k for each sample.

Sample	Flow Index n	Consistency Index K	r^2^
ML0	0.558 ± 0.002 ^c^	44.20 ± 0.44 ^a^	0.997
ML25	0.593 ± 0.002 ^a^	28.45 ± 0.11 ^c^	0.992
ML50	0.577 ± 0.004 ^b^	29.06 ± 0.30 ^c^	0.989
ML75	0.529 ± 0.004 ^d^	28.94 ± 0.63 ^c^	0.991
ML100	0.407 ± 0.009 ^e^	33.01 ± 1.14 ^b^	0.974

Superscript numbers with different letters within the same column are statistically different (*p* < 0.05). ML0: control sample with no oil substitution; ML25: sample with 25% WL and 75% oil; ML50: sample with 50% WL and 50% oil; ML75: sample with 75% WL and 25% oil; ML100: sample with 100% WL.

**Table 4 foods-12-02584-t004:** The nutritional contents of muffins with varying levels of fat substitution. Data are expressed as grams per 100 g of dry matter (DM), and the calorie values are provided as kilocalories per 100 g of fresh product.

Sample	ML0	ML25	ML50	ML75	ML100
Protein	6.72 ± 0.23 ^a^	6.34 ± 0.63 ^a^	6.67 ± 0.46 ^a^	6.67 ± 0.06 ^a^	6.96 ± 0.52 ^a^
Fat	13.14 ± 0.08 ^a^	9.80 ± 0.07 ^b^	6.59 ± 0.30 ^c^	4.68 ± 0.13 ^d^	1.82 ± 0.08 ^e^
Carbohydrates	55.21 ± 0.62 ^a^	54.64 ± 0.30 ^a^	54.51 ± 1.11 ^a^	54.12 ± 1.64 ^a^	56.38 ± 0.66 ^a^
Ash	1.36 ± 0.04 ^b^	1.35 ± 0.03 ^b^	1.44 ± 0.01 ^ab^	1.49 ± 0.01 ^ab^	1.54 ± 0.04 ^a^
Dietary Fibers	0.30 ± 0.08 ^d^	0.28 ± 0.00 ^d^	1.14 ± 0.00 ^c^	2.27 ± 0.32 ^b^	3.16 ±0.02 ^a^
Calories	366 ± 24 ^a^	332 ± 20 ^b^	300 ± 17 ^c^	285 ± 13 ^d^	270 ± 12 ^e^

Superscript numbers with different letters within the same line are statistically different (*p* < 0.05). ML0: control sample with no oil substitution; ML25: sample with 25% WL and 75% oil; ML50: sample with 50% WL and 50% oil; ML75: sample with 75% WL and 25% oil; ML100: sample with 100% WL.

**Table 5 foods-12-02584-t005:** Technological properties of muffins with different levels of substitution of fat. Data are expressed on a DM basis.

Sample	Height (cm)	Baking Loss (%)	Volume (cm^3^)	Density (g/cm^3^)	Moisture (g/100 g)
ML0	4.53 ± 0.10 ^a^	14.1 ± 1.0 ^a^	87.8 ± 5.7 ^b^	0.50 ± 0.04 ^b^	22.20 ± 0.71 ^d^
ML25	4.53 ± 0.08 ^a^	9.15 ± 0.93 ^bc^	95.7 ± 3.5 ^a^	0.48 ± 0.00 ^b^	28.25 ± 0.07 ^c^
ML50	4.48 ± 0.12 ^ab^	10.46 ± 0.78 ^b^	92.2 ±5.7 ^ab^	0.50 ± 0.03 ^b^	30.15 ± 0.49 ^bc^
ML75	4.43 ±0.19 ^ab^	7.1 ± 1.1 ^d^	73.9 ± 3.5 ^c^	0.63 ± 0.02 ^a^	31.90 ± 0.42 ^ab^
ML100	4.37 ± 0.03 ^b^	7.9 ± 1.2 ^dc^	73.5 ± 2.3 ^c^	0.63 ± 0.02 ^a^	32.95 ± 0.07 ^a^

Superscript numbers with different letters within the same column are statistically different (*p* < 0.05). ML0: control sample with no oil substitution; ML25: sample with 25% WL and 75% oil; ML50: sample with 50% WL and 50% oil; ML75: sample with 75% WL and 25% oil; ML100: sample with 100% WL.

**Table 6 foods-12-02584-t006:** Pore characterization of muffins’ crumbs with different levels of substitution of fat.

Sample	Area Fraction (%)	Pore Density (Pores/cm^2^)	Perimeter (mm)	Circularity (%)	D1,0	D3,2
ML0	10.76 ± 0.36 ^a^	12.5 ± 2.0 ^b^	2.84 ± 0.26 ^a^	0.81 ± 0.01 ^a^	1.41 ± 0.09 ^a^	4.02 ± 0.33 ^a^
ML25	10.91 ± 0.37 ^a^	24.5 ± 5.1 ^a^	2.13 ± 0.30 ^b^	0.81 ± 0.01 ^a^	1.13 ± 0.10 ^ab^	3.39 ± 0.60 ^b^
ML50	10.64 ± 0.30 ^a^	23.7 ± 4.2 ^a^	1.83 ± 0.26 ^c^	0.81 ± 0.01 ^a^	1.13 ± 0.10 ^b^	3.07 ± 0.31 ^b^
ML75	10.30 ± 0.20 ^a^	25.1 ± 3.9 ^a^	1.94 ± 0.23 ^c^	0.82 ± 0.01 ^a^	1.18 ± 0.09 ^b^	2.81 ± 0.23 ^b^
ML100	10.74 ± 0.29 ^a^	14.1 ± 2.6 ^b^	2.79 ± 0.27 ^a^	0.81 ± 0.01 ^a^	1.41 ± 0.07 ^a^	3.94 ± 0.43 ^a^

Superscript numbers with different letters within the same column are statistically different (*p* < 0.05). ML0: control sample with no oil substitution; ML25: sample with 25% WL and 75% oil; ML50: sample with 50% WL and 50% oil; ML75: sample with 75% WL and 25% oil; ML100: sample with 100% WL.

**Table 7 foods-12-02584-t007:** Cielab parameters of muffins with different levels of substitution of fat. Beneath, a picture of the cross-section of the experimental muffins.

Crumb/Crust	Cielab Parameters	ML0	ML25	ML50	ML75	ML100
Crumb	*L*	73.50 ± 0.95 ^a^	61.1 ± 5.6 ^b^	61.0 ± 2.4 ^b^	55.5 ± 3.0 ^c^	45.4 ± 3.4 ^d^
	*a**	−3.63 ± 0.37 ^e^	−1.03 ± 0.49 ^d^	3.29 ± 0.39 ^c^	5.44 ± 0.39 ^b^	7.9 ± 0.51 ^a^
	*b**	41.8 ± 3.1 ^a^	21.3 ± 2.2 ^b^	8.7 ± 1.39 ^c^	7.9 ± 1.2 ^c^	5.35 ± 0.23 ^d^
Crust	*L*	70.0 ± 1.6 ^a^	60.3 ± 6.0 ^b^	59.9 ± 3.3 ^b^	44.2 ± 2.4 ^c^	36.8 ± 3.6 ^d^
	*a**	−2.17 ± 0.50 ^e^	−1.33 ± 0.48 ^d^	2.39 ± 0.34 ^c^	5.80 ± 0.91 ^b^	7.46 ± 0.83 ^a^
	*b**	45.6 ± 3.4 ^a^	24.0 ± 1.3 ^b^	8.7 ± 1.2 ^c^	9.0 ± 1.4 ^cd^	7.0 ± 1.3 ^d^
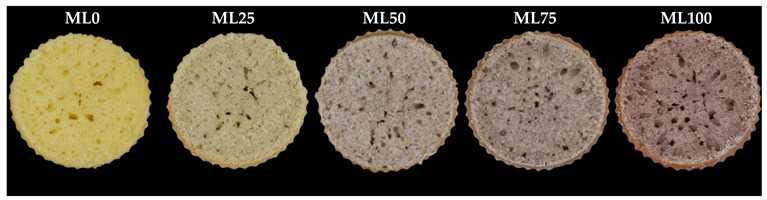						

Superscript numbers with different letters within the same line are statistically different (*p* < 0.05). ML0: control sample with no oil substitution; ML25: sample with 25% WL and 75% oil; ML50: sample with 50% WL and 50% oil; ML75: sample with 75% WL and 25% oil; ML100: sample with 100% WL.

**Table 8 foods-12-02584-t008:** Nutritional content of muffins with or without different levels of substitution of fat. Data are expressed on a DM basis.

Sample	Hardness (N)	Cohesiveness	Chewiness (N)	Resilience
ML0	38.0 ± 4.6 ^c^	0.52 ± 0.01 ^c^	19.8 ± 2.0 ^e^	0.32 ± 0.05 ^d^
ML25	35.3 ± 4.80 ^c^	0.64 ± 0.01 ^b^	29.0 ± 3.0 ^d^	0.40 ± 0.05 ^c^
ML50	46.4 ± 6.1 ^c^	0.63 ± 0.01 ^b^	32.43 ± 0.51 ^c^	0.42 ± 0.06 ^c^
ML75	73 ± 7.5 ^b^	0.71 ± 0.05 ^a^	49.7 ± 8.7 ^b^	0.52 ± 0.05 ^b^
ML100	99.1 ± 9.3 ^a^	0.72 ± 0.02 ^a^	67.8 ± 5.0 ^a^	0.59 ± 0.05 ^a^

Superscript numbers with different letters within the same column are statistically different (*p* < 0.05). ML0: control sample with no oil substitution; ML25: sample with 25% WL and 75% oil; ML50: sample with 50% WL and 50% oil; ML75: sample with 75% WL and 25% oil; ML100: sample with 100% WL.

## Data Availability

Data is contained within the article.

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
