# Peer review of "The Potential of Wine Lees as a Fat Substitute for Muffin Formulations"

_foods, 2023, doi:10.3390/foods12132584_

Round 1
Reviewer 1 Report
The authors have submitted a manuscript describing a study focused on replacement of fat in muffins with wine lees. This focus should appeal to those with an interest in nutrition as well as those with an interest in sustainability via valorisation of waste streams.
The manuscript will require significant attention to correct issues with ESL. These are primarily related to correct English spellings, grammar, and sentence structure. While I was generally able to understand what the authors were attempting to communicate, other readers with ESL may not be able to do so.
A few other points I would like to see addressed:
1) The introduction to the manuscript calls out yeast mannoproteins, yet these are never mentioned in the text as a potential modulator of batter rheology (Page 5). Batter rheology is discussed in terms of pectin content, which is not mentioned in the introduction. The authors should add a short discussion of pectins to their introduction and attempt to further the discussion of batter rheology by adding some information regarding mannoproteins.
2) In the discussion of muffin volume, the authors call out bubble coalescence and bubble loss as possible causes of lower volume in WL75 and WL100 (Page 7). However, given the batter rheology results, it is also possible that this is influenced by lack of gas bubble expansion.
3) The increase in muffin moisture content with increasing WL (Table 5) raises some concerns in my mind regarding potential shelf-life/microbial stability. The authors have not mentioned this anywhere in the document. Given the potential for wine lees to introduce additional microbial populations to the muffin and the fact that baking is not a kill step, I would think that a discussion of the moisture content and its relationship to shelf-life is warranted.
This manuscript will require significant attention to address ESL issues.
Reviewer 2 Report
The manuscript describes the influence of different concentrations of wine lees on some physicochemical properties of muffins. The manuscript has several problems and is not acceptable in its present form:
Comments:
Line 44: what are lipid-based fat replacers?! Do they have lower calorie compared to oils and fats?
Line 87: ventilated oven
Lines 146 and 147: In line 146 you have used "colour" but in line 147 you have written "color" please write similar words.
Table 3: The statistical analysis for flow index is not correct.
Table 5: The statistical analysis for "height" is not correct.
lines 278-179: Please rewrite these sentences. You can write "The color of crust is formed during baking operation due to the caramelization of sugars and Maillard reaction. Incorporation of wine lees increases the Maillard reaction by increasing the amino acid and carbohydrate content of muffins. While the crumb color is affected by cake ingredients and the pigments of wine lees increase the color parameters of muffins. You can use the following reference: 10.1016/j.lwt.2021.110914
Lines 303-304: These sentences are vaguely written; please paraphrase the sentences.
Sensory analysis: The sensory analysis is not correct. The scores should be base on the acceptability by consumers not the intensity of a parameter. For instance wine taste and odor are not positive properties in cake but samples with the highest wine taste has got higher scores. The dark color of cake crumb is a negative property but dark samples have received higher scores. You should report the acceptability not the intensity of color. Please correct the sensory parameters. Furthermore, how did you evaluate acidity by sensory analysis?!
The manuscript should be edited by a professional English editor.
